# From Automated Synthesis to In Vivo Application in Multiple Types of Cancer—Clinical Results with [^68^Ga]Ga-DATA^5m^.SA.FAPi

**DOI:** 10.3390/ph15081000

**Published:** 2022-08-14

**Authors:** Lukas Greifenstein, Carsten S. Kramer, Euy Sung Moon, Frank Rösch, Andre Klega, Christian Landvogt, Corinna Müller, Richard P. Baum

**Affiliations:** 1CURANOSTICUM Wiesbaden-Frankfurt, Center for Advanced Radiomolecular Precision Oncology, 65191 Wiesbaden, Germany; 2Department of Chemistry–TRIGA, Institute of Nuclear Chemistry, Johannes Gutenberg University, 55128 Mainz, Germany

**Keywords:** nuclear medicine, PET, Ga-68, FAP, FAPI, DATA, squaric acid, molecular imaging, chelator, first-in-human study

## Abstract

Radiolabeled FAPI (fibroblast activation protein inhibitors) recently gained attention as widely applicable imaging and potential therapeutic compounds targeting CAF (cancer-associated fibroblasts) or DAF (disease-associated fibroblasts in benign disorders). Moreover, the use of FAPI has distinct advantages compared to FDG (e.g., increased sensitivity in regions with high glucose metabolism, no need for fasting, and rapid imaging). In this study, we wanted to evaluate the radiochemical synthesis and the clinical properties of the new CAF-targeting tracer [^68^Ga]Ga-DATA^5m^.SA.FAPi. The compound consists of a (radio)chemically easy to use hybrid chelate DATA.SA, which can be labeled at low temperatures, making it an interesting molecule for ‘instant kit-type’ labeling, and a squaric acid moiety that provides distinct advantages for synthesis and radiolabeling. Our work demonstrates that automatic synthesis of the FAP inhibitor [^68^Ga]Ga-DATA^5m^.SA.FAPi is feasible and reproducible, providing convenient access to this new hybrid chelator-based tracer. Our studies demonstrated the diagnostic usability of [^68^Ga]Ga-DATA^5m^.SA.FAPi for the unambiguous detection of cancer-associated fibroblasts of various carcinomas and their metastases (NSCLC, liposarcoma, parotid tumors, prostate cancer, and pancreas adenocarcinoma), while physiological uptake in brain, liver, intestine, bone, and lungs was very low.

## 1. Introduction

Since targeted radiopharmaceuticals against certain cancer-related antigens (such as PSMA and SSTR) have demonstrated the clinical impact of radiomolecular precision oncology, the demand became apparent to find new targets with a suitable profile for targeted imaging and therapy. An ideal neoantigen should be highly expressed on malignant tissue and not only be restricted to a small subset of cancer types (pan-cancer target). The fibroblast activation protein (FAP) can be seen as a very interesting candidate as it is not only expressed on several tumor types, but also on cancer-associated fibroblasts (CAFs) that are part of the tumor microenvironment (TME) [1,2,3,4,5,6,7]. FAP is a transmembrane glycoprotein (type II) and belongs to the serine protease family [8,9]. The unique feature of FAP is that it is nearly absent in healthy tissues as well as in benign tumors. In contrast, FAP is overexpressed in over 90% of the most common human epithelial tumors, such as breast, pancreatic, colon, and lung cancers; it is also associated with conditions or diseases such as wound healing, chronic inflammation, liver cirrhosis, rheumatoid arthritis, pulmonary fibrosis, bone and soft tissue sarcomas, osteoarthritis, and Crohn’s disease [8,10,11,12,13,14,15,16,17,18,19,20,21,22,23,24,25,26,27,28,29,30]. For a recent review, see Imlimthan et al. [31].

Currently, most FAP-targeted radiopharmaceuticals are highly affine inhibitors of the FAP enzyme (FAPI, FAP inhibitors) that possess a low off-target selectivity towards other peptidases.

So far, the best known FAP inhibitors are boronic acid-based and glycine-2-cyanopyrrolidine-based FAPIs; for those, the boronic acid acts as the ‘warhead’, while the nitrile group in the (cyano)proline-derived FAPIs takes over this function (Figure 1A). In particular, the gly-(2-cyano)Pro inhibitors show very high selectivity for FAP and additionally sufficient non-specificity for competing peptidases [32,33,34]. The most potent inhibitor is (*S*)-N-(2-(2-cyano-4,4-difluoropyrrolidin-1-yl)-2-oxoethyl)quinoline-4-carboxamide (UAMC1110), developed by Pieter van der Veken et al. [35,36,37].

Accordingly, the first FAP-targeting inhibitor compounds that found application for radiomolecular precision imaging were DOTA-conjugated variants derived from UAMC1110 [38,39,40]. In preclinical studies, especially FAPI-04 and FAPI-46 conjugates gained attention as the most promising FAP-addressing radiopharmaceuticals to date (Figure 1B) [41,42,43]. Since then, FAPI derivatives found their way into several clinical applications for a wide variety of tumor types, primarily with a focus on diagnosis by ^68^Ga-PET/CT. However, increasingly, other PET nuclides, such as ^18^F and ^64^Cu [44,45,46,47,48,49,50,51,52,53,54], as well as beta and alpha emitters, are tested in preclinical studies, e.g., ^177^Lu [55,56,57,58,59,60], while therapeutical nuclides such as ^153^Sm [61], ^90^Y, ^177^Lu and ^225^Ac [62,63,64,65,66,67] and combinations thereof (“TANDEM” approach) have entered clinical phases [68].

In recent years, several groups have tried to design molecules that combine the advantages of both cyclic and acyclic chelators as so-called hybrid chelators. Hybrid chelators are generally considered to be particularly fast in their radiolabeling kinetics, and temperature independent when compared to acyclic chelators. Furthermore, kinetic stability should be comparable to cyclic complexes [69,70,71]. A very promising approach is the usage of a diazepine scaffold which contributes with two endocyclic amines to complex formation. Introduction of an additional exocyclic amine function provides a third coordination unit. Further functionalization of these amines with carboxylic acids allows the introduction of more donor units and final formation of the DATA (6-amino-1,4-diazepine-triacetic acid) chelator (Figure 1C) [72,73]. For the development of these structures to bifunctional systems, many derivatives were investigated. In this study, DATA^5m^ is used: herein, a C5-linker with a terminal carboxylic acid is introduced to provide bifunctionality (Figure 1C). In previous studies, it could be shown that DATA possess excellent properties for complexing gallium at room temperature resulting in highly stable complexes (note: even though reaction is possible at room temperature, we applied slightly elevated temperatures in our clinical routine work due to validated production methods).

In the present study, we used the DATA^5m^-conjugated precursor DATA^5m^.SA.FAPi, which was first introduced by Moon et al. [74] (Figure 1D). The synthesis of the precursor takes advantage of a squaric acid (SA) moiety that conveniently allows the coupling of two amine bearing units [75,76], such as the chelator and FAP-targeting motif. SA is not only acting as a connector, the influence of the SA unit on the pharmacophore and the biological activity of the final compound must be considered as well. Favorable kinetics by SA have been shown for other molecules before [77,78]. For the coupling of chelators, it is also noteworthy that SA has complexing properties itself [79] and may therefore influence the complexation properties of the bound chelator.

While the FAP-targeted and radiolabeled SA derivatives DOTA.SA.FAPi and AAZTA^5^.SA.FAPi have been investigated preclinically and clinically in several studies [74,80,81,82,83], there are only two clinical case reports by Kreppel et al. that investigated [^68^Ga]Ga-DATA^5m^.SA.FAPi in fourteen patients with focal nodular hyperplasia (FNH) (one patient) and neuroendocrine tumors (13 patients) [84,85]. In this study, we present our results on automated radiolabeling of DATA^5m^.SA.FAPi with Ga-68 and furthermore our first experience in patients with different cancers such as small-cell cancer of the prostate, parotid gland tumor, liposarcoma, pancreatic adenocarcinoma and non-small-cell lung cancer (NSCLC).

## 2. Results

### 2.1. Synthesis and Radiochemistry

Synthesis of the molecule was performed in accordance with the literature published by Moon et al. [74].

Radiolabeling of the new FAP inhibitor DATA^5m^.SA.FAPi with gallium-68 was performed automatically using a mini-All-in-One cassette-based module from Trasis. Reaction at 50 °C in sodium acetate buffer (0.7 M, pH 5.5) provided RCY ≥ 95% within 10 min. An example of the radio-HPLC and radio-TLC is shown in Figure 2, demonstrating a radiochemical purity (RCP) ≥ 98% (avg. RCP 99.28 ± 0.99%). The automated synthesis yielded activities between 616 and 654 MBq (avg. 634 ± 16 MBq; *n* = 4; radiochemical yield (RCY = 84.4 ± 5.0%) after 10 min of labeling with 50 μg of DATA^5m^.SA.FAPi and 20 min of quality control. 

No indices for tracer instability were detected in the period between synthesis and image acquisition. 

### 2.2. Clinical Safety

All patients tolerated the tracer application (median: 168 MBq, range: 126–316 MBq) extremely well without any acute adverse effects. No symptoms were noticed by any patient during the follow-up period of approx. 4 weeks (by consultation and clinical examination), and blood tests (CBC, liver, and kidney parameters). 

### 2.3. In-Human Tracer Uptake and Biodistribution

Mild physiological uptake was predominantly found in salivary glands and the thyroid, gall bladder (and consequently in intestines), frequently in the uterus and—due to excretion—in the kidneys and the urinary bladder. Pancreatic uptake was inhomogeneous and varied considerably between patients, with accentuation in the head and/or tail. Table 1 summarizes the physiological uptake (SUV_max_ and SUV_mean_) in selected tissues and organs (collected data from all examined patients). Figure 3 illustrates a patient with sarcoidosis that did not show any disease-related tracer uptake. As patients were not imaged with other FAPI tracers, data for direct comparison are not available. To benchmark [^68^Ga]Ga-DATA^5m^.SA.FAPi with other FAP-directed tracers, we compared the tracer distribution pattern of [^68^Ga]Ga-DATA^5m^.SA.FAPi with published data from FAPI-02, FAPI-46, and FAPI-74 at comparable time points [86,87]. At best, the physiological distribution of the squaric acid-based analog is similar to the pattern of FAPI-02 but with less kidney uptake and less blood pool activity. Blood pool and muscle uptake are significantly better compared to FAPI-46 and FAPI-74, while uptakes in the thyroid gland, glandular submandibularis, and pancreas are higher for [^68^Ga]Ga-DATA^5m^.SA.FAPi as well as for FAPI-02 (compared to FAPI-46 and FAPI-74). 

In contrast, malignant lesions showed a higher uptake of [^68^Ga]Ga-DATA^5m^.SA.FAPi with an average SUV_max_ of 9.1 ± 3.3, resulting in a mean tumor-to-background ratio (TBR) of 4.7. The highest SUV_max_ of 12.7 was found in a bone metastasis of a patient with prostate cancer, while the highest tumor uptake could be detected in a liver metastasis with a SUV_max_ of 5.0 in a parotid cancer patient. Notably, these observations are not statistically significant as only one patient for each tumor entity was examined. 

Figure 4A demonstrates tracer uptake in a patient with heavily metastatic parotid gland tumor: while uptake of the primary did not exceed physiological accumulation in the salivary glands, liver metastasis could be easily demarcated. In small-cell prostate cancer patients (Figure 4B), soft tissue and bone metastases could be visualized with high contrast and a remarkably high tumor SUV_max_ of 13.7. Additionally, liposarcoma-derived peritoneal metastases as well as a adenocarcinoma of the pancreas showed excellent tracer uptake with SUV_max_ values of approx. 10 (Figure 4C,D). 

In a case with metastasized NSCLC, the primary could be easily visualized (Figure 5) as well as the known cerebral metastases (Figure 5B left) (previously known from FDG-PET/CT that was performed 2 weeks before, Figure 5C,D). Notably, in [^68^Ga]Ga-DATA^5m^.SA.FAPi PET/CT, new bone lesions were detected that were previously not known (Figure 5B right).

Table 2 summarizes additional findings, clinical details as well as information regarding relevant pretreatments.

## 3. Discussion

The automatic synthesis allows a fast, easy and reproducible preparation of [^68^Ga]Ga-DATA^5m^.SA.FAPi at low temperatures and in high RCPs and RCYs. These favorable conditions allow the ‘kit-type synthesis’ of this tracer, making it a very convenient imaging agent for FAP—even for PET centers that possess only basic technical equipment and infrastructure.

Currently, only two clinical case reports with [^68^Ga]Ga-DATA^5m^.SA.FAPi, focusing on FNH and on metastasized neuroendocrine tumors, were published [84]. Our study confirms the clinical safety of this compound as all patients tolerated the application of the tracer very well. Moreover, we demonstrate its first application in patients with a diversity of solid tumors.

The radiopharmaceutical demonstrated excellent diagnostic usability for the unambiguous detection of CAFs of various carcinomas (such as NSCLC, prostate, and pancreas) as well as in liposarcoma and in soft tissue and bone metastases. High tumor uptake was observed, and very low accumulation in brain, liver, bone, and lungs. In the patient with a pancreas head carcinoma (Figure 4D), the primary tumor could be undoubtfully visualized despite some physiological uptake in the pancreas. In line with the findings of Kreppel et al. [84], liver metastases could also readily be detected with this tracer. A clear advantage of FAP-targeting tracers in comparison to FDG is that metastases are not masked by high physiological FDG uptake. Therefore, imaging with [^68^Ga]Ga-DATA^5m^.SA.FAPi allowed also visualization of cerebral metastasis (Figure 5B left). Furthermore, bone lesions that were visible on [^68^Ga]Ga-DATA^5m^.SA.FAPi-PET/CT had not been detected by FDG-PET/CT. With regard to physiological biodistribution, the here presented tracer has a similar distribution pattern to FAPI-02 but with less kidney uptake and less blood pool activity, while radiolabeling can be achieved under more suitable conditions. As this study shows only one patient per cancer entity, tumor uptakes cannot be compared in a statistically significant manner. 

A limitation of our study is the heterogenous patient group and small sample size, making a general statement about sensitivity and specificity in specific malignancies difficult. Furthermore, a bigger sample size is necessary to allow comparisons between DATA^5m^.SA.FAPi, other FAP-targeting tracers, and FDG-PET. Additionally, the potential of the tracer to image disease-associated fibroblasts (DAFs) needs to be further explored: although Kreppel et al. [84,85] revealed that [^68^Ga]Ga-DATA^5m^.SA.FAPi-PET/CT visualizes FNH (a benign disease), no active lesion was seen in a patient with a past history of sarcoidosis (Figure 3). During the time of the PET scan, the activity status was unclear while the patient suffered from several seizures; therefore, the missing tracer uptake cannot be correlated with disease (in) activity. 

In conclusion, our results demonstrate that the FAP-targeting tracer [^68^Ga]Ga-DATA^5m^.SA.FAPi can conveniently be synthesized in an automatized process and that malignant lesions of solid tumors could be detected in a non-preselected heterogenous patient population. At this point, further studies need to be performed to compare the sensitivity of this molecule with established imaging agents. As FAP-imaging has clear advantages over FDG imaging, the synthetic accessibility to [^68^Ga]Ga-DATA^5m^.SA.FAPi, along with the excellent first clinical experience, can provide a good opportunity even for PET centers with basic equipment to implement this tracer in daily routine.

## 4. Materials and Methods

### 4.1. Radiochemistry

Gallium-68 was eluted from a ^68^Ge/^68^Ga generator (Eckard und Ziegler AG, Braunschweig, Germany) and used without further purification. Radiolabeling was performed in 2.0 mL of 0.7 M ammonium acetate buffer at pH 5.5 with 50 µg precursor at 50 °C in an All-In-One mini synthesis module (Trasis, Ans, Belgium) with a disposable cassette. Subsequently, an Oasis HLB Plus cartridge was used to agitate the final compound. Elution of the product was performed with 0.5 mL ethanol. The product was then formulated via a sterile filter with 10 mL of saline. 

The pH was controlled at the start and after the labeling. For reaction control, radio-TLC (TLC Silica gel 60 F254 Merck) with (1) citrate buffer pH 4 and (2) a 1:1 mixture of ammonium acetate buffer and MeOH (*v*:*v*) as the mobile phase was used. Additionally, radio-HPLC using an analytical HPLC from Agilent (Infinity 1200) with a Ramona * radio-detector (Elysisa-Raytest, Angleur, Belgium) (Column: VDSpher PUR 150 C18-E 5 μm, 100 × 40 m), linear gradient of 5–45% MeCN (+0.1% TFA)/95–55% Water (+0.1% TFA) in 10 min). TLCs were measured with a TLC miniGita (Elysia-Raytest, Angleur, Belgium) with the analysis software GINA (Elysia-Raytest, Angleur, Belgium).

### 4.2. Clinical

Five patients had histologically confirmed malignant disease, while one patient was diagnosed with sarcoidosis. All patients had an ECOG performance status <3, good renal function, and no allergies against any of the ingredients of the radiopharmaceutical. All patients signed written informed consent before this study.

Tracer uptake and biodistribution were analyzed in 6 patients (mean age: 56 y; range: 49–63 y) with various malignancies: small-cell cancer of the prostate (male, 63 y), malignant parotid gland tumor (adenoid-cystic subtype) (female, 54 y), liposarcoma (male, 62 y), pancreatic adenocarcinoma (female, 54 y), and NSCLC (female, 51 y). A sixth patient had a history of sarcoidosis (with no information about active lesions during the time of the scan) (female, 49 y).

After injection of [^68^Ga]Ga-DATA^5m^.SA.FAPi (median: 168 MBq, range: 126–316 MBq), PET/CT images (Siemens Biograph Vision 600 Edge) were acquired (malignant patients: from the vertex to mid-thigh; sarcoidosis patient: from skull to mid-thigh) after an average uptake time of 70 min (range: 48–83 min). A low-dose CT scan (120 keV, 30 mAs, CareDose; reconstructed with a soft-tissue kernel to a slice thickness of 2 mm) was used for attenuation correction and anatomical mapping of the tracer. An infusion of 500 mL saline with 20 mg of furosemide was infused from 15 min before to 150 min after tracer application.

Tumor uptake was quantified by SUV_mean_ and SUV_max_. SUV_max_ data were calculated on the reconstructed images by drawing circular regions of interests around various organs and tumor lesions using Affinity Viewer (Hermes Medical Solutions). The tumor-to-background ratio (TBR) was defined as SUV_max_ of the target lesions, and the SUV_mean_ of healthy liver parenchyma. 

Blood samples were drawn before tracer application (CBC, liver, and kidney parameters) if no recent blood test results were available. During application, patients were monitored by medical personnel and proactively asked about side effects. Patients that were not seen in a follow-up period of 4 weeks were requested to contact us if side effects appeared. Blood results after the scan were acquired in our center or inquired from the patient.

## 5. Patents

L.G. and F.R. have filed a patent (WO2020083853A1) on [^68^Ga]Ga-DATA^5m^.SA.FAPi.

## Figures and Tables

**Figure 1 pharmaceuticals-15-01000-f001:**
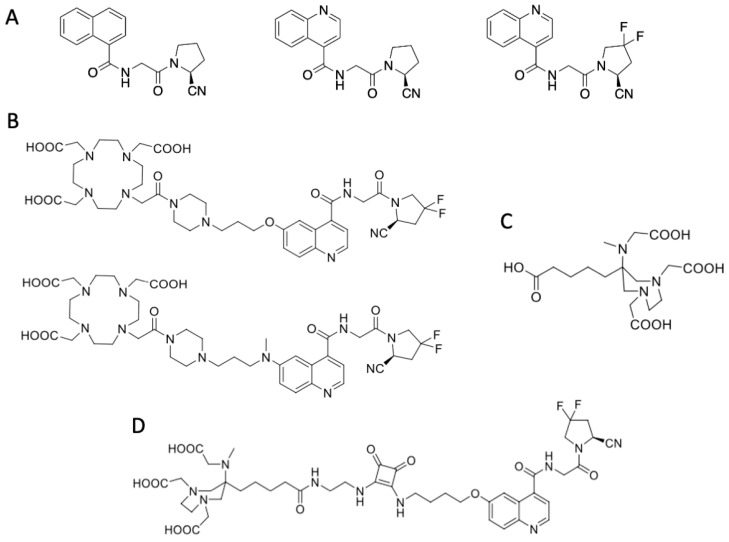
Schematic representation of the relevant chemical structures. (**A**): Gly-2-cyanopyrrolidine FAP inhibitors: (N-(1-naphthyl)-gly-2-cyanopyrrolidine; (4-quinolinyl)glycyl-2-cyanopyrrolidine and the quinoline-gly-2-cyano-4,4-difluoroPro-based FAPi, UAMC1110; (**B**) DOTA-conjugated FAPI conjugates FAPI-04 (top) and FAPI-46 (bottom); (**C**) DATA^5m^ (**D**) DATA^5m^. SA.FAPi.

**Figure 2 pharmaceuticals-15-01000-f002:**
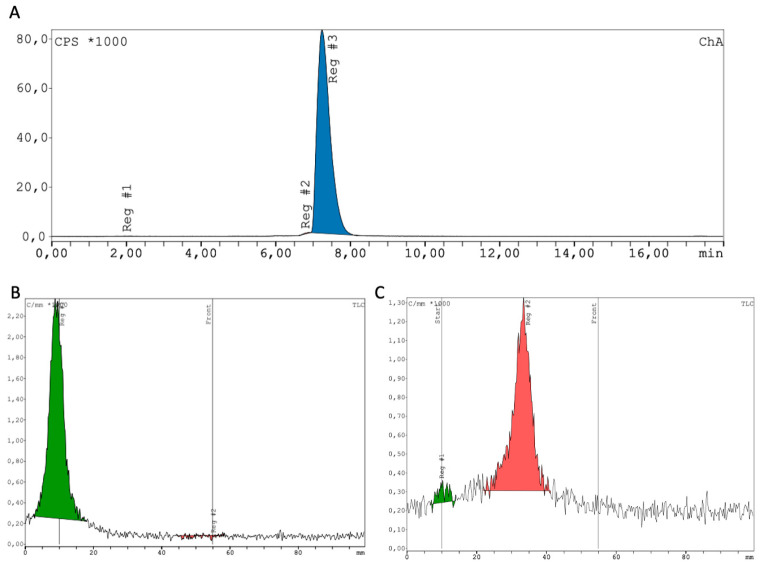
Quality control of [^68^Ga]Ga-DATA^5m^.SA.FAPi. (**A**) radio-HPLC; (**B**) radio-TLC in citric acid buffer; (**C**) radio-TLC in ammoniumacetate buffer and methanol (1:1 *v*/*v*).

**Figure 3 pharmaceuticals-15-01000-f003:**
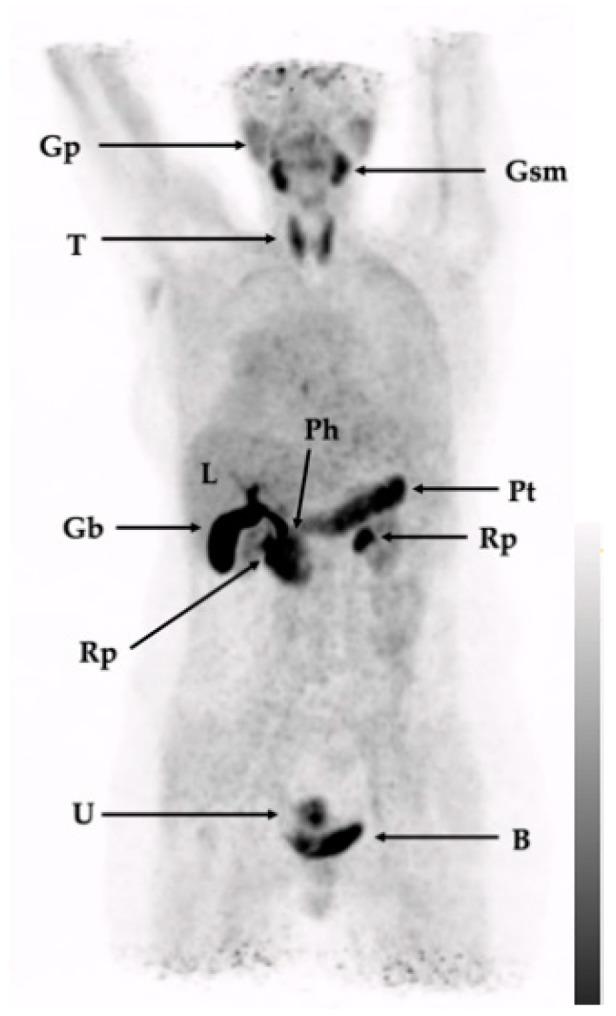
Maximum-intensity projection (MIP) of [^68^Ga]Ga-DATA^5m^.SA.FAPi in a patient without any disease-related tumor uptake. Gp: glandula parotis; Gsm: glandula submandibularis; T: thyroid; L: liver; Ph: pancreas head; Pt: pancreas tail; Gb: gall bladder; Rp: renal pelvis; U: uterus; B: urinary bladder.

**Figure 4 pharmaceuticals-15-01000-f004:**
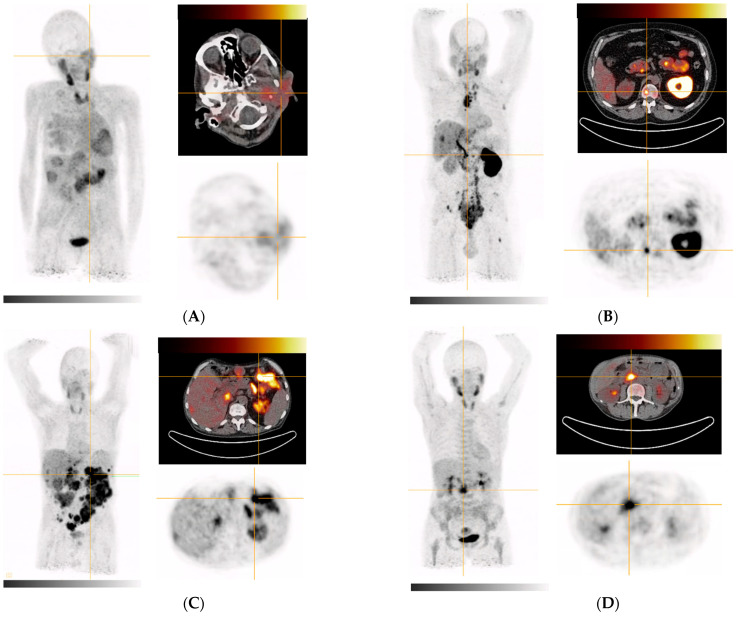
MIPs and transversal images of [^68^Ga]Ga-DATA^5m^.SA.FAPi in cancer patients. Crosshair indicates target lesion (highest tumor SUV_max_); (**A**) hepatic metastases of parotid gland tumor; target lesion (primary tumor/liver metastasis): 5.0; (**B**) metastasized prostate cancer; target lesion (bone metastasis in L1 vertebra): 12.7; (**C**) metastasized liposarcoma; target lesion (peritoneal metastases): 10.6; (**D**) primary pancreatic head adenocarcinoma tumor: 10.1.

**Figure 5 pharmaceuticals-15-01000-f005:**
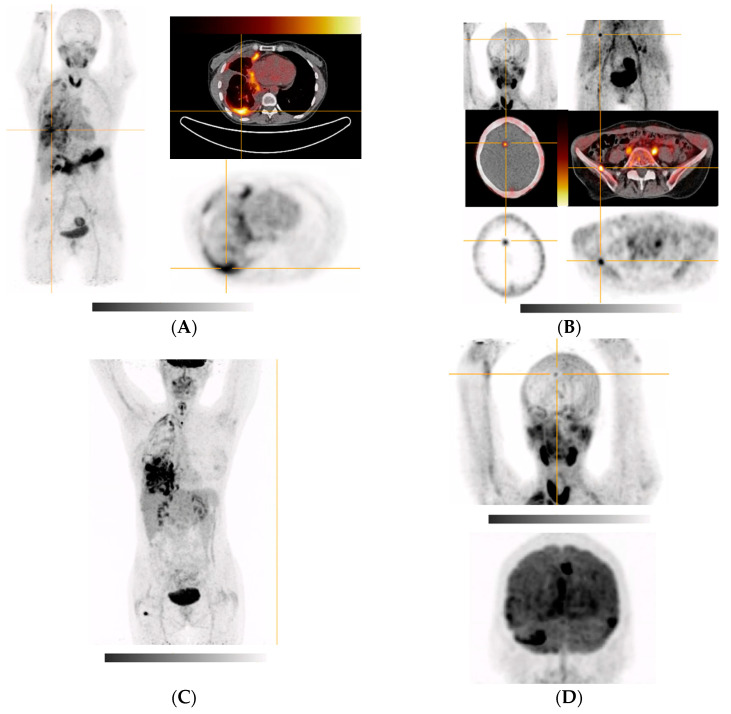
(**A**) MIP and transversal image of [^68^Ga]Ga-DATA^5m^.SA.FAPi in a NSCLC patient. Crosshair indicates target lesion with a SUV_max_ of 7.3. (**B**) Cerebral (**left**) and bone (**right**) metastases demonstrated by FAPI-PET/CT. (**C**,**D**) MIPs, FDG scan performed two weeks before imaging with [^68^Ga]Ga-DATA^5m^.SA.FAPi; the upper picture in (**D**) is the enlarged image of the head shown in (**B**).

**Table 1 pharmaceuticals-15-01000-t001:** Physiological uptake (SUV_max_ and SUV_mean_) of the tracer [average ± SD] in selected organs and tissues.

Tissue/Organ	SUVmax [Average ± SD]	SUVmean [Average ± SD]
Spleen	2.4 ± 0.5	1.6 ± 0.5
Liver	2.5 ± 0.4	1.9 ± 0.3
Red marrow (vertebrae)	2.3 ± 0.4	1.5 ± 0.5
Kidney	3.0 ± 1.1	2.4 ± 0.9
Brain (frontal cortex)	0.4 ± 0.2	0.1 ± 0.1
Pituitary gland	1.7 ± 0.6	1.0 ± 0.3
Submandibular gland	6.3 ± 1.2	4.3 ± 2.1
Thyroid	6.1 ± 1.8	3.9 ± 1.2
Pancreas	7.8 ± 2.5	4.5 ± 1.5
Lung	0.8 ± 0.3	0.6 ± 0.2
Muscle (quadriceps)	0.9 ± 0.2	0.6 ± 0.2
Blood pool (aorta)	1.8 ± 0.9	1.5 ± 0.7

**Table 2 pharmaceuticals-15-01000-t002:** Findings in all six patients during imaging with [^68^Ga]Ga-DATA^5m^.SA.FAPi.

Patient	Disease	Pretreatments(Relevant)	Tracer Accumulation
1 (Figure 3)	Sarcoidosis (left cervical lymph node manifestation), in the past inflammatory pulmonary activity, the activity of the sarcoidosis during image acquisition was not clear. History of DCIS in the left breast	Lymph node extirpation, modified radical mastectomy on left side, prophylactic radical modified mastectomy on right side	Only physiological tracer distribution with no further suspect tracer accumulations
2 (Figure 4A)	Metastasized parotid gland tumor (adenoid-cystic subtype)	Subtotal parotidectomy (left), partial liver resection of segments II/III	Parotid gland, multiple masses in the right lung, accumulation in several liver segments
3 (Figure 4B)	Metastasized prostate cancer (Gleason 4 + 3 = 7)	IMRT to prostate and seminal vesicles	Extensive bone and bone marrow involvement (including the extremities), large lymph node metastases in the retroperitoneum, mediastinum (bulky), and cervical region as well as in soft tissue. Additional findings: Severely impaired function of left kidney (slow wash out of activity with high parenchymal contrast and increased activity in the ureter and pelvis)
4 (Figure 4C)	Metastasized liposarcoma	Tumor enucleation on the duodenum, pancreatic head resection, omentectomy, pancreaticogastrostomy	Extensive peritoneal tumor foci in all abdominal quadrants, caudal liver margin segment VI (or adjacent peritoneal foci)
5 (Figure 4D)	Primary pancreatic head adenocarcinoma, moderately differentiated, ductal	ERCP with stenting	Inhomogeneous in the pancreas with emphasis in pancreatic head, peritoneal (extensively in the left mid to lower abdomen), anterior margin of liver (peritoneal or lymph nodes), mammaria interna lymph node, segment VI of the liver (most likely biliary excreted tracer), uterus (nonspecific/physiological), muscle attachments at the hip joints (nonspecific/inflammatory/bursitis)
6 (Figure 5A,B)	Poorly differentiated NSCLC of right lower lobe (pleural, cerebral, hepatic, and osseous metastases)	VATS with partial pleurectomy and talc pleurodesis	Right hemithorax and right lung, diffuse cerebral (mainly focal in the right frontal cortex in the region of the great longitudinal fissure and periventricular), right and right ilium, left sacrum, right thigh (after surgical removal of a hibernoma), uterus (possible myoma) FDG-PET (Figure 5C,D): whole right hemithorax (especially basolateral, in the myelon, intracranially in the area of the meninges, (around the temporal poles and on the tentorium), in the right frontal cortex in the area of the great longitudinal fissure, right thigh (after surgical removement of a hibernoma)

## Data Availability

The data presented in this study are available in the article.

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
