# Peer review of "From Automated Synthesis to In Vivo Application in Multiple Types of Cancer—Clinical Results with [68Ga]Ga-DATA5m.SA.FAPi"

_pharmaceuticals, 2022, doi:10.3390/ph15081000_

Round 1
Reviewer 1 Report
In the present study, Greifenstein et al. have shown that the automated radiosynthesis of [68Ga]DATA5m.SA.FAPI can be produced in a simple manner, rapidly and with high quality for patient use, using a mini all-in-one module from Trasis. This potentially offers the possibility for nuclear medicine facilities to produce the radiotracer in a standardised and safe way. FAP-directed radioligands have proven to be very valuable for non-invasive imaging in recent years, as the radioligands developed for this purpose have a very good signal-to-background ratio and are very specific. Due to the small number of patients, a statistical analysis has only been carried out to a limited extent, however it is not absolutely necessary for this type of study. In turn, patients with different malignancies were selected, which once again underlines the broad range of application and the enormous benefit for the clinical diagnosis/therapy of FAP-targeted radioligands. In this respect, the present study is consequential and recommended for acceptance in principle on my behalf, after the following points have been addressed:
minor:
TBR - does this describe tumour-to-background or tumour-to-blood ratio (line 237 vs. 138), please correct.
major:
Only SUVmax values are given, please add the SUVmean and correlate the values to the previous clinical studies
In general I could not find any values regarding stability/metabolism of the radioligand in blood/plasma of patients neither in this nor in the previous studies, please add if possible
With regard to pancreatic uptake, please comment for the association between diabetes and FAP (inhibitors) and, if possible, include the patient status (diabetes type or normal).
Thank you very much for your work!
Reviewer 2 Report
Greifenstein and colleagues, herein, present their results on the automated synthesis and clinical application of [68Ga]Ga-DATA5m.SA.FAPi for PET/CT imaging of sarcoidosis and different cancers in six patients. The authors show the feasibility of the automated synthesis of this tracer with excellent radiochemical yield and radiochemical purity. Application in humans shows uptake in tumors with a good target-to-background ratio.
This is an interesting work coming from two groups with a track record of first-in-human studies in radionuclide imaging and therapy. There a couple of limitations of this work, though. Below are a few suggestions for improvement.
Specific comments
- Title: In view of the prior work of Kreppel et al. reporting the in-human use of [68Ga]Ga-DATA5m.SA.FAPi, I doubt if it is correct to call this report a “first in vivo application”. I suggest that the authors edit the title in this regard.
- Methods: “All patients had histologically confirmed malignant disease,…” In view of one patient with sarcoidosis who was included in this study, this statement may not be entirely correct. Please correct.
- Methods: “…acquired from the vertex to mid-thigh after an average uptake time of 70 min (range: 48–83 min).” Again, the image of the patient with sarcoidosis shown in figure 3 was acquired from the base of the skull to mid-thigh and not vertex to mid-thigh. Please correct.
- Clinical safety: This section provides results regarding the tolerability of [68Ga]Ga-DATA5m.SA.FAPi in the patients. Please provide a description of the steps taken to collect these pieces of safety information including the schedule of patients’ visits, blood workup done, and others in the Methods section.
- Figure 4A: Please specify the histological subtype of the parotid tumor this patient has.
- Figures 3, 4, and 5: Please include color scale bars for all images.
- Please include information detailing the clinical, pathologic and, imaging findings for each patient. In this table, it may be relevant to mention findings on other imaging modalities acquired for the clinical care of the patients. This may provide readers an opportunity to compare the performance of this tracer against imaging modalities performed as part of the standard of care in these tumor types.
- Discussion: “Furthermore, bone lesions that were visible on [68Ga]Ga-DATA5m.SA.FAPi-PET/CT had not been detected by FDG- PET/CT.” More information is required here regarding this discordance in imaging findings. For example, it is necessary to state the time interval between the FDG and FAPI PET/CT scans. Also, it may be worth it to include the FDG PET/CT images along with the FAPI images in figure 5 to give the readers the opportunity to compare images side-by-side between the two scans. Here, in the discussion, it should be emphasized that while FAPI may have shown an additional lesion in the spine, this does not influence management in any way, especially in this patient with metastatic NSCLC.
- Discussion: Please expand further on the discussion. It will be necessary to compare the performance of this tracer reported here and other FAPI tracers out there, especially with respect to uptake in tumors and normal organ physiologic distribution. This will be necessary to show the superiority or otherwise of this tracer compared with other FAP-targeting radiopharmaceuticals.
- Discussion: Please emphasize what you think the potential role of this tracer may be in the clinical management of oncologic patients.
- Limitations: “…although Kreppel et al. (86, 87) revealed that [68Ga]Ga-DATA5m.SA.FAPi-PET/CT visualizes FNH (a benign disease), no active lesion was seen in a patient with a past history of sarcoidosis (Fig 3).” But does this patient has disease at the time of FAPi imaging? What are the findings on imaging acquired for the clinical care of this patient? That there are no tracer-avid lesions seen in this patient is not sufficient to conclude that this tracer is not taken up in benign lesions. This could mean that the patient is in remission and has no active disease at the time of the index FAPi imaging. In fact, FAP expression has been well-documented in various benign conditions. Therefore, the lack of uptake of this tracer in known benign lesions would be concerning.
- Conclusion: “…that malignant lesions of solid tumors could be detected with high sensitivity in a non-preselected heterogenous patient population.” I will like to caution the authors in concluding regarding this tracer’s sensitivity in the absence of data showing the relative performance of this FAP-targeting PET/CT imaging versus validated imaging techniques acquired as part of the clinical care of these patients.
Reviewer 3 Report
Dear the authors.
I am really grateful to be given the opportunity to review the article, entitled "From automated synthesis to first in vivo application - clinical results with [68Ga]Ga-DATA5m.SA.FAPi in different types of cancer".
The result has shown that malignant lesions of solid tumors could be detected with high sensitivity in a non-preselected heterogenous patient population by using FAP-targeting tracer, which could lead to the suggestion that FAP-imaging has clear advantages over FDG 198
imaging.
The conclusion should be really worthwhile to be shared among clinicians.
I have one comment shown below, which, I might think, should be addressed for possible publication.
Comment;
The order of the submitted manuscript has been arranged as follows;
Introduction, Results, Discussion, Materials and Methods, and Patents.
Materials and Methods section should be placed before the result section.
I hope my comment would be helpful for the authors to improve their study presentation.
Thank you.
Round 2
Reviewer 1 Report
Dear Authors,
thank you very much for improving your manuscript and to answer the questions as good as possible. Therefore, I have nothing to state and recommend your manuscript for publication.
Just one recommendation to you or the editors:
Could you please check the formatting of the tables, especially table 2, before publishing? It is not easy to read.
Author Response
Thank you